# LEARNABLE GROUP TRANSFORM FOR TIME-SERIES

## ABSTRACT

We undertake the problem of representation learning for time-series by considering a Group Transform approach. This framework allows us to, first, generalize classical time-frequency transformations such as the Wavelet Transform, and second, to enable the learnability of the representation. While the creation of the Wavelet Transform filter-bank relies on affine transformations of a mother filter, our approach allows for non-linear transformations. This is achieved by sampling a subset of invertible maps on $\mathbb{R}$. The subset considered contains strictly increasing and continuous functions. The transformations induced by such maps enable us to span a larger class of signal representations, from wavelet to chirplet-like filters. We propose a parameterization of such a non-linear map such that its sampling can be optimized for a specific loss and signal. The Learnable Group Transform can thus be cast into a Deep Neural Network. The experiments on diverse time-series datasets demonstrate the expressivity of this framework, which competes with state-of-the-art performances.

## 1 INTRODUCTION

The selection of the time-frequency representation for analyzing, classifying, and predicting time-series has long been studied (Coifman & Wickerhauser, 1992; Mallat & Zhang, 1993; Gribonval & Bacry, 2003). To this day, the front-end processing of time-series remains a keystone toward the improvement of a wealth of applications such as health-care (Saritha et al., 2008)), environmental sound (Balestriero et al., 2018; Lelandais & Glotin, 2008), and seismic data analysis (Seydoux et al., 2016). The common denominator of the recorded signals in these fields is their undulatory behavior. While these signals share this common behavior, two significant factors imply the need of learning the representation: **1)** time-series are intrinsically different because of their physical nature, **2)** the machine learning task can be different even within the same type of data. Therefore, the representation should be induced by both the signal and the task at hand.

An all too common approach to performing inference on time-series consists of building a Deep Neural Network (DNN) that operates on a spectral decomposition of the time-series such as Wavelet Transform (WT) or Mel Frequency Spectral Coefficients (MFSC). The selection of the judicious transform is either performed by an expert in the signal at hand, or by considering the aforecited selection methods and their derivatives. However, an inherent drawback is that the selection of the time-frequency transform is often achieved with criteria that do not align with the task. For instance, a selection based on the sparsity of the representation while the task is the classification of the signals. Besides, these selection methods and transformations require substantial cross-validations of a large number of hyperparameters such as mother filter family, number of octaves, number of wavelets per octave, size of the window (Cosentino et al., 2017).

To alleviate these drawbacks, Ravanelli & Bengio (2018); Balestriero et al. (2018); Cakir et al. (2016); Zeghidour et al. (2018) investigated the learnability of a mother filter. This learnable mother filter is transformed by deterministic affine maps. These transformations constitute the filter-bank. The representation of the signal is obtained by convolving the filter-bank atoms with the signals. Recently, Khan & Yener (2018) investigated the learnability of the affine transformations, that is, the sampling of the dilation parameter of the affine group inducing the wavelet filter-bank. Optimized jointly with the DNN, their method allows an adaptive transformation of the mother filter. Another approach consists of building equivariant-invariant representations. In Mallat (2012); Bruna (2013) they propose a translation-invariant representation, the Scattering Transform, which is stable under

the action of small diffeomorphisms. In Oyallon et al. (2018); Cohen & Welling (2016), they focus on equivariant-invariant representations for images, which reduces the sample complexity and endow DNN's layers with interpretability.

In this work, we focus on GT, which is achieved by taking the inner product between the filter-bank, which is built by taking the action of a transformation map on a mother filter, and the signal. Well-known GTs are the Short-Time Fourier Transform (STFT) and the Continuous Wavelet Transform (CWT). We propose to extend these GTs and improve their flexibility by introducing the Learnable Group Transform (LGT) by **1)** generalizing the affine transformations of a mother filter leading to wavelet filter-bank by introducing non-linear transformations (Section 3.1), **2)** proposing a parameterization of such non-linear map such that it can be learned efficiently and jointly with any DNN, (Sections 3.2, 3.3), **3)** Replacing the affine transformations of CWT by non-linear maps allows for greater flexibility in the learnable spectral decomposition which displays different equivariance properties (Section 3.4). This flexibility improves the linearization capability of the representation as it eases the learning of a spectral decomposition that is able to discard intricate patterns in the time-series that are nuisances. This specific transformation of a filter induces a filter with a non-linear instantaneous phase, which in turn, allows to span filters a la chirplets, which are of interest in a variety of domains such as biology and medicine, mechanics and vibrations, and sonar systems (Flandrin, 2001). Also, this generalization implies that for fixed network topology, replacing the learnable affine group with the continuous group leads to a larger class of spannable functions, which improves the approximation property of the DNN at hand (Winkler & Le, 2017; Balestriero & Baraniuk, 2018). In order to show the generality of our approach, we apply our algorithm on two diverse time-series classification problems (Section 4).

## 2 BACKGROUND AND NOTATIONS

We first highlight the properties of particular GTs by expressing their time-frequency tiling.

### 2.1 TIME-FREQUENCY TILING

The spread of a filter and its Fourier transform are inversely proportional as per the Heisenberg uncertainty principle (Mallat, 1999). Following this principle, we can observe that in the case of STFT (respectively WT with a Gabor wavelet), at a given time $\tau$, the signal is transformed by a window of constant bandwidth (respectively proportional bandwidth) modulated by complex exponential resulting in a uniform tiling (respectively proportional) on the frequency axis, Figure (1). In the case of a chirp-like filter, as proposed in Baraniuk & Jones (1996), each tile is a sheared rectangular, more generally, an affinely transformed rectangular. In this case, as well,

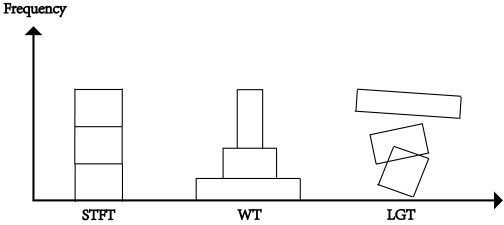

Figure 1: **Time-Frequency Tilings** at a given time $\tau$: (*left*) Short-Time Fourier Transform, i.e., constant bandwidth, (*middle*) Wavelet Transform, i.e., proportional bandwidth, (*right*) Learnable Group Transform, i.e, adaptive bandwidth, the "tiling" is induced by the learned non-linear transformation underlying the filter-bank decomposition.

the lower bound area of the sheared rectangular is constrained by the uncertainty principle. As such, the understanding of the benefits of various time-frequency decompositions can be achieved by analyzing how they tile the time-frequency plane. For instance, in the case of WT, the precision in frequency degrades as the frequency increases while its precision in time increases (Mallat, 1999). In the case of STFT, the uniform tiling implies that the precision is constant along the frequency axis. In our proposed framework, the LGT allows for an adaptive tiling, as illustrated in Figure (1) such that the trade-off between time and frequency precision depends on the task and data.

## 3 LEARNABLE GROUP TRANSFORM

To extend the filter-bank derivation as proposed in a wavelet decomposition we introduce a learnable group transform. We now define a subset of invertible maps on $\mathbb{R}$ enabling the transformation of a mother filter . Then, we provide a parameterization of such functions and show how one can efficiently learn these parameters. Finally, we derive the equivariance properties of the induced group transform. The overall building block the LGT and its application on a signal is depicted in Figure (2).

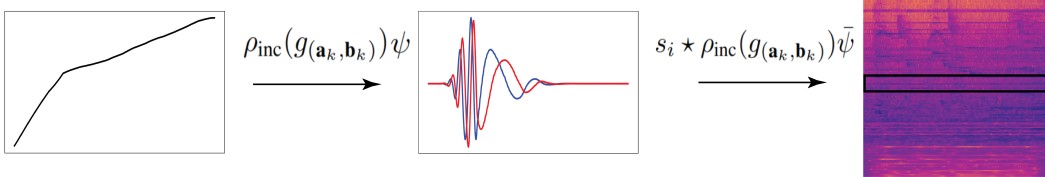

Figure 2: **Learnable Group Transform:** (left) generating the strictly increasing continuous functions $\rho_{\text{inc}}(g_{(\mathbf{a}_k, \mathbf{b}_k)})$ which stands for the strictly increasing and continuous transformation operator with parameters $(\mathbf{a}_k, \mathbf{b}_k)$, $\forall k \in \{1, \ldots, K\}$, where $K$ denotes the number of filters in the filter-bank. (middle) Each generated operators $\rho_{\text{inc}}(g_{(\mathbf{a}_k, \mathbf{b}_k)})$ are applied to the mother filter denoted by $\psi$ (presently a Morlet wavelet), where the imaginary part is shown in red and the real part in blue. This transformation leads to the filter-bank, $\rho_{\text{inc}}(g_{(\mathbf{a}_k, \mathbf{b}_k)})\psi$ where $g_{(\mathbf{a}_k, \mathbf{b}_k)} \in C_{\text{inc}}(\mathbb{R})$. Then, the convolution between this generated filter-bank and the signal leads to the LGT of the signal. The black box on the LGT representation (right) corresponds to the convolution of the $k^{\text{th}}$ filter with the signal. The strictly increasing and continuous piece-wise linear functions can be learned efficiently by back-propagating the error induced by the generated GT.

### 3.1 STRICTLY INCREASING AND CONTINUOUS TRANSFORMATIONS

In order to generalize the classical affine transformations used in WT, we propose the utilization of strictly increasing and continuous functions defined as

$$C_{\text{inc}}(\mathbb{R}) = \{g \in C(\mathbb{R}) | g \text{ is strictly increasing}\}, \tag{1}$$

where $C(\mathbb{R})$ defines the space of continuous functions defined on $\mathbb{R}$. This set of function is composed of invertible maps which is crucial in order to derive invariance properties as well as avoid artifacts in the transformed filters.

We define the linear operator $\rho_{\text{inc}}(g)$ by

$$[\rho_{\text{inc}}(g)\psi](t) = \psi(g(t)), \quad \forall \psi \in \mathbb{L}_2(\mathbb{R}), \forall g \in C_{\text{inc}}(\mathbb{R}), \tag{2}$$

where $\psi$ denotes a mother filter. We can see that the increasing and continuous group representation operator $\rho_{\text{inc}}$ induces a mapping which depends on the function $g \in C_{\text{inc}}(\mathbb{R})$. If for instance $g = e$ , i.e., the identity map, then we have $\rho_{\text{inc}}(e)\psi = \psi$, it is in fact the identity operator in the space of the mother filter. Given a mother filter $\psi \in \mathbb{L}_2(\mathbb{R}), \rho_{\text{inc}}(g)\psi, \forall g \in C_{\text{inc}}(\mathbb{R})$ induces a non-linear transformation of the mother filter which can be visualized in Figure (3). Note that in signal processing, such a transformation is called warping (Goldenstein & Gomes, 1999; Kerkyacharian et al., 2004).

| $g \in C_{inc}(\mathbb{R})$ | $\psi(g(t))$ |
|---|---|
| Affine | Wavelet |
| Quadratic Convex | Increasing Quadratic Chirplet |
| Quadratic Concave | Decreasing Quadratic Chirplet |
| Logarithmic | Logarithimic Chirplet |
| Exponential | Exponential Chirplet |

Table 1: Special cases of the function $g$ inducing filters belonging to well-known filter-banks.

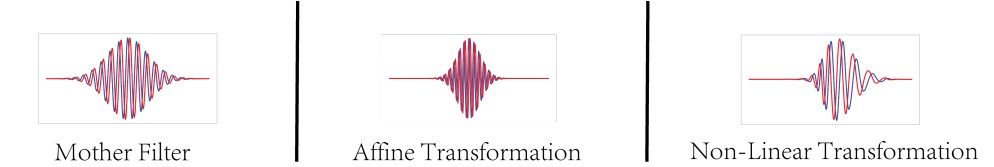

Figure 3: **Transformation of a Morlet Wavelet:** For all the filters, the real part is shown in blue and the imaginary in red. (*left*) Morlet wavelet mother filter. (*middle*) Transformation of the mother filter with respect to an affine transform: the dilation parameter $0 < a < 1$, i.e., contraction, and translation $b = 0$, i.e., no translation. (*right*) Increasing and continuous transformation of the mother filter for some randomly generated function $g \in C_{\text{inc}}(\mathbb{R})$ leading to chirplet-like filter.

Among the possible transformations induced on a mother filter by the mapping $g \in C_{\text{inc}}(\mathbb{R})$, some of them correspond to well-known filters (Table 1).

## 3.2 Sampling the Transformation Maps

In this work, we are specifically interested in the learnability of such an increasing and continuous map. As such, we provide a way to sample such a space via its parameterization. We propose to use piece-wise affine functions constrained such that they belong to the class of strictly increasing and continuous functions. This constrained piece-wise affine map is defined as

$$g_{(\mathbf{a},\mathbf{b})}(t) = \sum_{l=1}^{n}(a_l t + b_l)\mathbf{1}_{I_l}(t), \quad \forall t \in \mathbb{R}, \tag{3}$$

$$\text{s.t.: } a_l > 0, \quad \forall l \in \{1,\ldots,n\}, \tag{4}$$

$$b_{l+1} = (a_l - a_{l+1})t_{l+1} + b_l, \quad \forall l \in \{1,\ldots,n-1\}, \tag{5}$$

where $\mathbf{a} = (a_1,\ldots,a_n)$, $\mathbf{b} = (b_1,\ldots,b_n)$, $\mathbf{1}_{I_l}$ is the indicator function of the intervals $I_l = [t_l, t_{l+1}), \forall l \in \{2,\ldots,n-1\}$ and $I_1 = (-\infty, t_1), I_n = [t_n, +\infty)$, and $a_l$ and $b_l$ denote respectively the slope and offset of each piece of the function and $n$ is the number of pieces. As such, for each $(\mathbf{a},\mathbf{b})$ satisfying the constraints (4) and (5) the function $g_{(\mathbf{a},\mathbf{b})}$ is a sample from the set $C_{\text{inc}}(\mathbb{R})$.

Notice that this mapping can be performed using a 1-layer ReLU Neural Network (Arora et al., 2016). This implementation implies a knot-free piece-wise affine mapping, providing more flexibility regarding the transformation map. The knot-free mapping is defined such that the uniform support, i.e., the intervals $I_l$ (3), is replaced with varying support for different $l \in \{1,\ldots,n\}$. As such, this flexibility induces better approximation property (Jupp, 1978).

## 3.3 Learning the Piece-wise Affine Transformation Maps

The parameters $(\mathbf{a}_k, \mathbf{b}_k), \forall k \in \{1,\ldots,K\}$ are differentiable with respect to the filter and thus any deep learning pipeline using those filters can be used and optimized jointly with the other DNN parameters by stochastic gradient descend methods. Given a set of signals $\{s_i \in \mathbb{L}_2(\mathbb{R})\}_{i=1}^{N}$ and given a task specific loss function $L$, we aim at solving the following optimization problem

$$\min_{(\mathbf{a}_1,\mathbf{b}_1)\in\Omega_1,\ldots,(\mathbf{a}_K,\mathbf{b}_K)\in\Omega_K} \sum_{i=1}^{N} L\big(F(\mathcal{W}[s_i,\psi](\mathbf{g},.)\big), \tag{6}$$

where $N$ denotes the number of signals, $K$ the number of filters, $F$ represents a DNN, $\Omega_k = \{\mathbf{a}_k \in \mathbb{R}_+^n, \mathbf{b}_k \in \mathbb{R}^n | \mathbf{b}_{(k,l+1)} = (\mathbf{a}_{(k,l)} - \mathbf{a}_{(k,l+1)})t_{(l+1,k)} + \mathbf{b}_{(k,l)}\} \forall k \in \{1,\ldots,K\}$, and $\mathcal{W}[s_i,\psi](\mathbf{g},.) = [\mathcal{W}[s_i,\psi](g_{(\mathbf{a}_1,\mathbf{b}_1)},.),\ldots,\mathcal{W}[s_i,\psi](g_{(\mathbf{a}_K,\mathbf{b}_K)},.)]^T$, and

$$\mathcal{W}[s_i,\psi](g_{(\mathbf{a}_k,\mathbf{b}_k)},.) = (s_i \star \rho_{\text{inc}}(g_{(\mathbf{a}_k,\mathbf{b}_k)})\bar{\psi})(.), \; g_{(\mathbf{a}_k,\mathbf{b}_k)} \in C_{\text{inc}}(\mathbb{R}), \; \forall k \in \{1,\ldots,K\}, \tag{7}$$

where $\bar{\psi}(t) = \psi(-t)$ and $(.)$ corresponds to the time axis.

We propose different settings that will impact the type of filter-bank our method can reach.

First, We propose a normalization of the frequency of the transform filter (denoted in the result tables by nLGT). This normalization helps to reduce the aliasing induced by the filters. We propose to use $\hat{f}$, the normalized frequency $f$ with respect to the maximum slope of the piece-wise affine mapping. For instance, in the case of a Morlet wavelet, the normalization is as follows

$$[\rho_{\text{inc}}(g_{(\mathbf{a},\mathbf{b})})\psi](t) = \pi^{-\frac{1}{4}} \exp\left(2\pi j \hat{f} g_{(\mathbf{a},\mathbf{b})}(t)\right) \exp\left(-\frac{1}{2}(g_{(\mathbf{a},\mathbf{b})}(t)/\sigma)^2\right),$$

where $\hat{f} = f/\max_{l \in \{1,\dots,n\}} a_l$, $j$ is the imaginary unit, and $\sigma$ is the width parameter defining the localization of the wavelet in time and frequency. This normalization will be performed for each sample of the group, and thus for each generated filter $k \in \{1, \dots, K\}$ of the filter-bank.

Second, we constrain the domain of the piece-wise affine map, as derived in (3) (denoted in the result tables by cLGT). In the following experiments, we propose a dyadic constraint of the domain as in the WT. The support of the filter is close to the support of a wavelet filter-bank. However, the envelope of the filter and the instantaneous frequency still vary as in the Chirplet Transform (Baraniuk & Jones, 1996).

### 3.4 EQUIVARIANCE PROPERTIES

The equivariance-invariance properties of signal representations play a crucial role in the efficiency of the algorithm at hand (Mallat, 2016). By considering the mapping $\rho_{\text{inc}}$ as a group action on the space of the mother filter, i.e., $\mathbb{L}_2(\mathbb{R})$, or more precisely, a representation of a group on $\mathbb{L}_2(\mathbb{R})$ we can develop the equivariance properties of the LGT. More details regarding the background of this group theoretical approach are given in Appendix A. We can consider the set $C_{\text{inc}}(\mathbb{R})$ with the operation $\odot$ consisting of the composition of function to form the group of strictly increasing and continuous maps denoted by $\mathbf{G}_{\text{inc}}$. This formulation eases the derivation of the equivariance properties of group transforms which can be defined for a group $\mathbf{G}$ by

$$\mathcal{W}[\rho(g')s_i, \psi](g, .) = \mathcal{W}[s_i, \psi]((g')^{-1} \odot g, .), \forall g, g' \in \mathbf{G}. \tag{8}$$

That is, transforming the signal with respect to the group $\mathbf{G}$ and computing its representation is equal to computing the representation of the signal and then transforming the representation. If $\mathbf{G}$ corresponds to the affine group, the associated group transform is the WT the transformation which is equivariant to scalings and translations. One can already notice that since $\mathcal{W}(.,.)$ employs convolution, for any group $\mathbf{G}$, the LGT is translation equivariant. We now focus on more specific equivariance properties of the LGT by defining the local equivariance by

$$\exists \tau \in \mathbb{R}, \mathcal{W}[\rho(g')s_i, \psi](g, \tau) = \mathcal{W}[s_i, \psi]((g')^{-1} \odot g, \tau), \forall g, g' \in \mathbf{G}.$$

That is, the representation of a local transformation of a signal in a window centred at $\tau$ is equals to the transformation of the representation at $\tau$. The size of the window depends on the support of the filter. As a matter of fact, assuming that the representation of $\mathbf{G}_{\text{inc}}$ is unitary, we have the following proposition.

**Proposition 1.** *The LGT is locally equivariant with respect to the action of the group $\mathbf{G}_{inc}$.*

Refer to Appendix E for the proof.

## 4 EXPERIMENTS

For all the experiments and all the settings, i.e., LGT, nLGT, cLGT, cnLGT, the increasing and continuous piece-wise affine map is initialized randomly, and the optimization is performed with Adam Optimizer, and the number of knots of each piece-wise affine map is 256. The mother filter used for our setting is a Morlet wavelet filter. The code of the LGT framework will be provided on the Github page of the first author.

### 4.1 ARTIFICIAL DATA: CLASSIFICATION OF CHIRP SIGNALS

We present an artificial dataset that demonstrates how a specific time-frequency tiling might not be adapted or would require cross-validations for a given task and data. To build the dataset, we

generate one high frequency ascending chirp and one descending high-frequency chirp of size $8192$ following the chirplet formula provided in (Baraniuk & Jones (1996)). Then for both chirp signals, we add Gaussian noise samples ($100$ times for each class), see Figures in Appendix (C.1). The task aims at being able to detect whether the chirp is ascending or descending. Both the training and test sets are composed of $50$ instances of each class. For all models, set the batch size to $10$, the number of epochs to $50$. Each experiment was repeated $5$ times with randomly sampled train and test set, and the accuracy was the result of the average over these $5$ runs. Each GT is composed with a non-linearity, and the inference is performed by a linear classifier. For the case of WT and LGT, the size of the filters is $512$. As we can observe in Table (2), the WT, as well as the STFT with few numbers of filters, perform poorly on this dataset. The chirp signals to be analyzed are localized close to the Nyquist frequency, and in the case of WT, as illustrated in Figure 1, the wavelet filter-bank has a poor frequency resolution in high frequency while benefiting from a high time resolution. In this experiment, we can see that this characteristic the WT time-frequency tiling implies that through time, the small frequency variations of the chirp are not efficiently captured.

In the case of STFT, as the number of filter decreases, the frequency resolution gets altered. Thus, this frequency variation is not captured. Using a large window for the STFT increases the frequency resolution of the tiling and thus enables to capture the difference between the two classes. In the LGT setting, the tiling has adapted to the task and produces good performances except for the cLGT model. In fact,

| Representation + Non-Linearity + Linear Classifier | Accuracy |
|---|---|
| Wavelet Transform (64 Filters) | $53.01 \pm 5.1$ |
| Short-Time Fourier Transform (64 Filters) | $65.1 \pm 11.9$ |
| Short-Time Fourier Transform (128 Filters) | $86.6 \pm 9.8$ |
| Short-Time Fourier Transform (512 Filters) | $\mathbf{100 \pm 0.0}$ |
| LGT (64 Filters) | $92.9 \pm 4.0$ |
| nLGT (64 Filters) | $95.7 \pm 3.3$ |
| cLGT (64 Filters) | $56.8 \pm 1.6$ |
| cnLGT (64 Filters) | $\mathbf{100.0 \pm 0.0}$ |

Table 2: Testing Accuracy for the Chirp Signals Classification Task

the domain of the piece-wise linear map is constrained to be dyadic, and thus the adaptivity of the filter bank is reduced, which is not suitable for this specific task. For all settings, the visualization of the filters, as well as the representations of the signals, can be found in Appendix (C.1.2,C.1.3). This experiment shows an example of signals that are not easily classified by neither the proportional-bandwidth nor the constant-bandwidth without considering cross-validation of hyperparameters.

## 4.2 Supervised Bird Detection

| Representation + Non-Linearity + Deep Network | AUC |
|---|---|
| MFSC (80 Filters) | $77.83 \pm 1.34$ |
| Conv. Filter init. random (80 Filters) | $66.77 \pm 1.04$ |
| Conv. Filter init. Gabor (80 Filters) | $67.67 \pm 0.98$ |
| Spline Conv. init. random (80 Filters) (Balestriero et al. (2018)) | $78.17 \pm 1.48$ |
| Spline Conv. init. Gabor (80 Filters) (Balestriero et al. (2018)) | $79.32 \pm 1.52$ |
| LGT (80 Filters) | $78.41 \pm 1.38$ |
| nLGT (80 Filters) | $75.50 \pm 1.39$ |
| cLGT (80 Filters) | $79.14 \pm 0.83$ |
| cnLGT (80 Filters) | $\mathbf{79.68 \pm 1.35}$ |

Table 3: Testing AUC for the Bird Detection Task

We now propose a large scale dataset to validate the suitability of our model in a noisy and realistic setting. The dataset is extracted from the Freesound audio archive Stowell & Plumbley (2013). This dataset contains about $7,000$ field recording signals of $10$ seconds sampled at $44$ kHz, representing slightly less than $20$ hours of audio signals. The content of these recordings varies from water sounds to city noises. Among these signals, some contain bird songs that are mixed with different background sounds having more energy than the bird song, see Appendix (C.2.1). The given task is a binary classification where one should predict the presence or absence of a bird song. As the dataset is unbalanced, we use the Area Under Curve (AUC) metric. The results we propose for both the benchmarks and our models are evaluated on a test set consisting of $33\%$ of the total dataset. In order

to compare with previously used methods, we use the same seeds to sample the train and test set, the batch size, i.e., 10, and the learning rate cross-validation grid as in Balestriero et al. (2018). For each model, the best hyperparameters are selected, and we train and evaluated randomly 10-times the models with early stopping, the results are shown in Table (3). While the first layer of the architecture has a model-dependent representation (i.e., MFSC, LGT, Conv. filters,...), we use the state-of-the-art architecture (Grill & Schlüter (2017)) for the DNN architecture, described in Appendix (B.2). Notice that this specific DNN architecture has been designed and optimized for MFSC representation. As we can see in Table 3, the case without constraints (LGT) reaches better accuracy than the domain expert benchmark (MFSC), showing the ability of such transformation to tile the time-frequency plane according to the task and data at hand. Besides, including more constraints on the model (cnLGT) reduces overfitting and further improve results to outperform the other benchmarks.

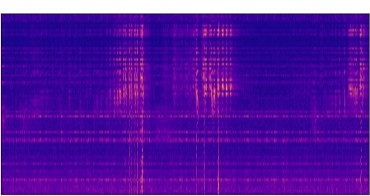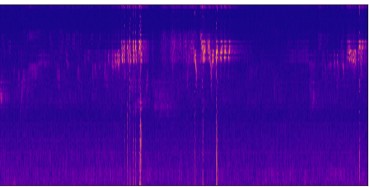

Figure 4: **Learnable Group Transform** - Visualisation of a sample containing a bird song (cLGT), where (*left*) at the initialization and (*right*) after learning. For each subfigure, the $x$-axis corresponds to time and the $y$-axis to the different filters. Notice that the $y$-axis usually corresponds to the scale or the center-frequency of the filters. Other representations are displayed in Appendix (C.2.3). We can observe that compared to the initialization, the learned representation is sparser and the SNR is increased. Besides, the representation is less redundant in the frequency axis.

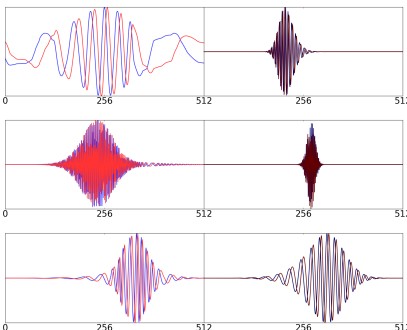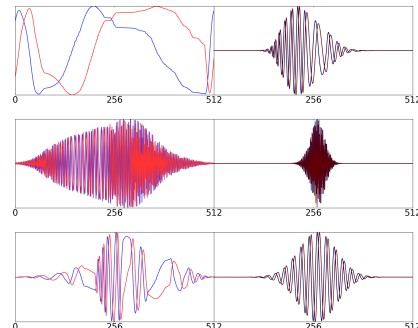

Figure 5: **Learnable Group Transform Filters** for the Bird Detection Data - Each row displays two selected filters (left and right sub-figure) for different settings: (*from top to bottom*) LGT, nLGT, cLGT. For each subfigure, the left part corresponds to the filter before training and the right part to the filter after training. The blue and red denote respectively the real and imaginary part of the filters.

One can notice that all the learned filters in Figure 5 contain either an increasing chirp or a decreasing chirp, corresponding respectively to the convexity or concavity of the instantaneous phase of the filter and thus of the piece-wise linear map. Such a feature is being used and is crucial in the detection and analysis of bird song (Stowell & Plumbley, 2012).

## 4.3 HAPTICS DATASET CLASSIFICATION

The Haptics dataset is a classification problem with five classes and 155 training and 308 testing samples from the UCR Time Series Repository Chen et al. (2015), where each time-series has 1092 time samples. As opposed to the bird dataset where features of interests are known, and competitive methods have been established, there is no expert knowledge regarding the specific signal features (see Table 4).

One can see that our method outperforms other approaches in the cLGT setting while performing the classification with a linear classifier as opposed to other methods using DNN algorithms. This demonstrates the capability of our method to transform the data efficiently while not requiring a further change of basis. Besides, even in a small dataset setting, our approach is capable of learning an efficient transformation of the data. We provide in

| Representation + Classifier | Accuracy |
|---|---|
| DTW (Al-Naymat et al. (2009)) | 37.7 |
| BOSS (Schäfer (2015)) | 46.4 |
| Residual NN (Wang et al. (2017)) | 50.5 |
| COTE (Bagnall et al. (2015)) | 51.2 |
| Fully Convolutional NN (Wang et al. (2017)) | 55.1 |
| WD + Convolutional NN (Khan & Yener (2018)) | 57.5 |
| LGT (96 Filters)+ Non-Linearity + Linear Classifier | 53.5 |
| nLGT (96 Filters)+ Non-Linearity + Linear Classifier | 50.4 |
| cLGT (96 Filters)+ Non-Linearity + Linear Classifier | **58.2** |
| cnLGT (96 Filters)+ Non-Linearity + Linear Classifier | 54.3 |

Table 4: Testing Accuracy for the Haptics Classification Task

Figure 6 the visualization of some sampled filters for each setting of the LGT model. As opposed to the supervised bird dataset, we can see that the filters do not coincide with well-known filters that are commonly used in signal processing. This is an example of an application where the features of interest in the signals are unknown, and one requires a learnable representation.

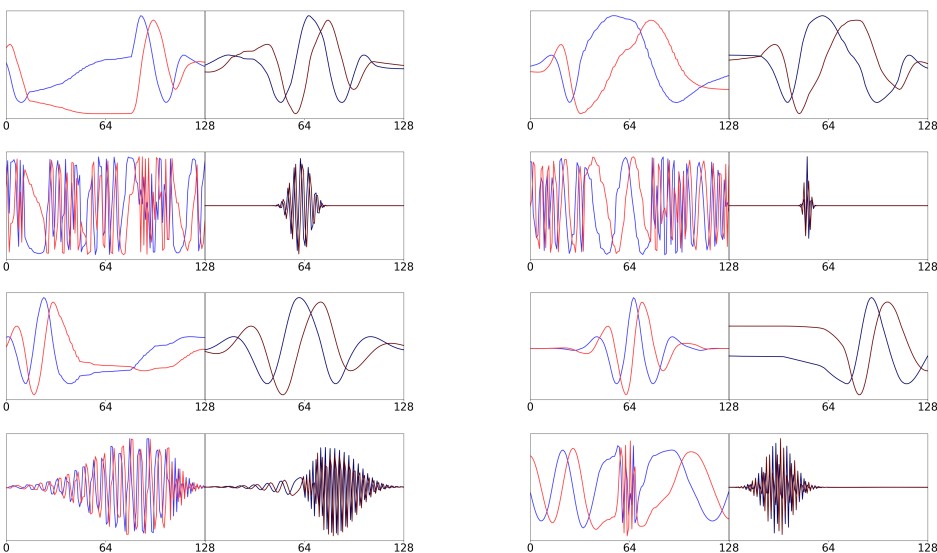

Figure 6: **Learnable Group Transform Filters** for the Haptics Data - Each row displays two selected filters (left and right sub-figure) for different settings: (*from top to bottom*) LGT, nLGT, cLGT, cnLGT. For each subfigure, the left part corresponds to the filter before training and the right part to the filter after training. The blue and red denote respectively the real and imaginary part of the filters.

## 5 CONCLUSION

We proposed to extend the WT by introducing a GT based on non-linear transformations of the mother filter. We restrain the transformation to be in the space of strictly increasing and continuous functions enabling its connection with well known time-frequency filters as well as the derivation of equivariance properties. We also shown a tractable way to learn to sample these transformations. From bird detection to haptics classification, our approach competes with state-of-the-art methods without a priori knowledge on the signal power spectrum and outperform classical hand-crafted time-frequency representations. Interestingly, in the bird detection experiment, we recover chirplet filters that are known to be crucial to their detection, while in the case of the haptic dataset where important features to be captured to perform the classification of the signals accurately are unknown, the filters learned are very dissimilar to classical time-frequency filters.

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

## A  GROUP TRANSFORM: A GROUP REPRESENTATION APPROACH

### A.1  BACKGROUND

For further details on the group theoretical aspects described in this section, the reader should refer to Vilenkin (1978).

**Definition 1.** *A group is a set $G$ with a multiplicative operation $\odot$ that respects enclosure, identity element, inverse element, and associativity.*

The representation of the group determines its action on a function space and bridges the gap between group theory and linear algebra, allowing to compute the transformation of a function following the rules induced by the specific group at hand. The representation of a group can be thought as a far-reaching generalization of the exponential function property, $\exp(x + y) = \exp(x)\exp(y), \forall x, y \in \mathbb{R}$ (Baraniuk, 1993). In fact, it is defined as,

**Definition 2.** *A linear continuous representation $\rho$ of a group $G$ on the linear space $\mathbb{H}$ is defined as*

$$\rho : G \rightarrow GL(\mathbb{H}), \tag{9}$$

*where $GL(\mathbb{H})$ is the the group of linear map in $\mathbb{H}$ such that $\forall g, g^{'} \in G$*

$$\rho(g \odot g') = \rho(g)\rho(g'). \tag{10}$$

For instance, let $\mathbb{H}$ be a vector space such as $\mathbb{R}^3$, the representation of the group is induced by $3 \times 3$ matrices. In this case, the operation on the right of (10) is a matrix multiplication, where each matrix depends on the group elements $g$ and $g'$. This concept extends to linear operators acting on functional spaces.

As such, multiple transformations of a function by different elements of the group is equal to the representation of the combination of the group elements applied to the function.

This structure-preserving map defines the action of a group on elements of function spaces. Group transforms such as STFT and CWT can be expressed in such a way by selecting a mother filter space and a group. The representation of the group in the mother filter space provides an operator that takes as input an element of the group and acts on the filter to transform it. A filter-bank can thus be created by iterating this process with different group elements. Therefore, the selected group carries the characteristics of the filter-bank and consequently, the group transform and its time-frequency tiling. The building blocks of the WT through representation theory is provided in Appendix A.2. Notice that further properties such as the invariant measure of the group and the resolution of the identity can be develop using the representation of the group.

### A.2  EXAMPLE: THE WAVELET TRANSFORM

As an introductory example, we consider the creation of a wavelet filter-bank utilizing transformation group. Let's denote by $\mathbf{G}_{\text{aff}}$ the affine group, the so called "$ax + b$" group, where the elements $(\lambda, \tau) \in \mathbb{R}_+^\star \times \mathbb{R}$, where $\mathbb{R}_+^\star = (0, +\infty)$, where the multiplicative operation of the group $\odot$ is defined by

$$(\lambda, \tau) \odot (\lambda', \tau') = (\lambda\lambda', \tau + \lambda\tau') \tag{11}$$

Let's define by $\rho_{\text{aff}}$ the representation of the affine group in $\mathbb{L}_2(\mathbb{R})$, i.e., $\rho_{\text{aff}} : \mathbf{G}_{\text{aff}} \rightarrow GL(\mathbb{L}_2(\mathbb{R}))$, such that $\rho_{\text{aff}}$ is a homomorphism as per Definition 2. Its action on square integrable function $\psi$ is defined as

$$[\rho_{\text{aff}}(g)\psi](t) = \frac{1}{\sqrt{\lambda}}\psi(\frac{t - \tau}{\lambda}), \ t \in \mathbb{R}, \tag{12}$$

where $(a, b)$ are respectively the dilation and translation parameters. The wavelet filter-bank is built by transforming a mother filter, $\psi$ by the representation $\rho_{\text{aff}}$ for specific elements of the group. A visualization of this approach for a Morlet wavelet filter can be seen in Figure (3). The wavelet transform of a signal $s_i \in \mathbb{L}_2(\mathbb{R})$ is achieved by

$$\mathcal{W}(s_i, \psi)(g_{(\lambda, \tau)}) = \left\langle s_i, \rho_{\text{aff}}(g_{(\lambda, \tau)})\psi \right\rangle, \forall g_{(\lambda, \tau)} \in \mathbf{G}_{\text{aff}}, \tag{13}$$

$$= (s_i \star \rho_{\text{aff}}(g_{(\lambda, 0)})\bar{\psi}), \forall g_{(\lambda, 0)} \in \mathbf{G}_{\text{aff}}, \tag{14}$$

where $\bar{\psi}_{(}t) = \psi(-t)$, $\langle .,. \rangle$ denotes the inner product, $\star$ the convolution, and $\rho_{\text{aff}}(g_{(\lambda,\tau)})\psi$ the action of the operator $\rho_{\text{aff}}$, evaluated at the group element $g_{(\lambda,\tau)}$, on the mother filter $\psi$ as per (12). In practice, the filter-bank is generated by sampling a few elements of the group. For instance, in the case of the dyadic wavelet transform, the dilation parameters follow a geometric progression of common ratio equals to 2. In general, the translation parameter is sampled according to the scaling one (Daubechies, 1992). Notice that in the convolution expression (14), the translation parameter $\tau = 0$, in fact the convolution operator $\star$ acts as the translation one. In the case where the translation parameter depends on the scaling one, a specific stride is used to perform the discrete convolution.

Note that the STFT can be constructed similarly utilizing the Weyl-Heisenberg group (Feichtinger et al., 2009), whose representation on $\mathbb{L}_2(\mathbb{R})$ consists of frequency modulations and translations. More intricated group representations can be built as in Torrésani (1991) where the combination of the affine group and Weyl-Heisenberg group is considered.

# B  Architecture Details

## B.1  Artificial Data



Group Transform + Complex Modulus + Log
Dense Layer (1 sigmoid)



After the Group Transform, a batch-normalization is applied.

## B.2  Supervised Bird Detection



Group Transform + Complex Modulus + Log + Average-Pooling (stride:$(1, 512)$ size:$(1, 1024)$)
Conv2D. layer (16 filters $3 \times 3$) and Max-Pooling ($3 \times 3$) and ReLU
Conv2D. layer (16 filters $3 \times 3$) and Max-Pooling ($3 \times 3$) and ReLU
Conv2D. layer (16 filters $3 \times 1$) and Max-Pooling ($3 \times 1$) and ReLU
Conv2D. layer (16 filters $3 \times 1$) and Max-Pooling ($3 \times 1$) and ReLU
Dense layer (256) and ReLU
Dense layer (32) and ReLU
Dense layer (1 sigmoid)



At each layer a batch-normalization is applied and for the last three layers a $50\%$ dropout is applied as in (Grill & Schlüter (2017)). The dimension of the input of the DNN presented is the same for the different benchmarks.

## B.3  Haptics Data



Group Transform + Complex Modulus + Log + Average-Pooling (stride:$(1, 64)$ size:$(1, 128)$)
Dense Layer (5 softmax)



After the Group Transform, a batch-normalization is applied.

## C ADDITIONAL FIGURES

### C.1 ARTIFICIAL DATA

#### C.1.1 DATA

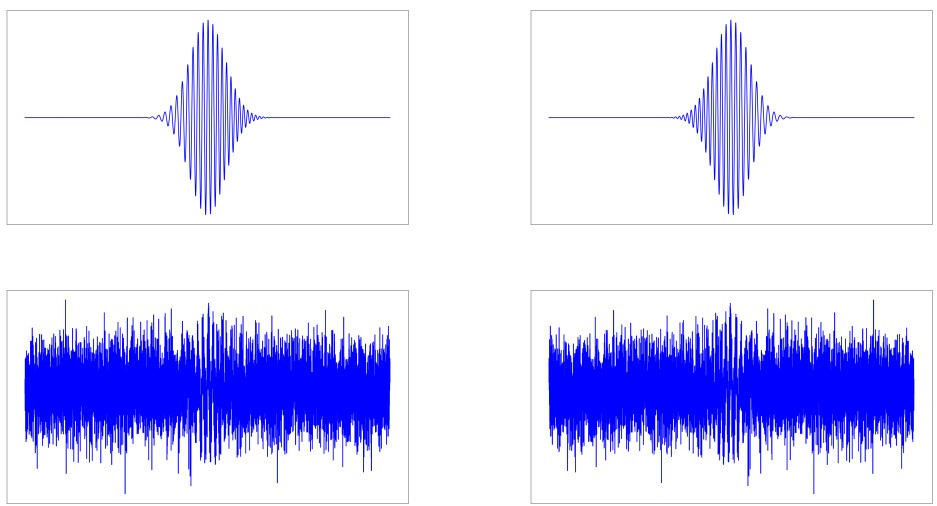

Figure 7: **Artificial Dataset**: (*Top Left*) Ascending Chirp, (*Top Right*) Descending Chirp, i.e. class 0, (*Bottom Left*) Ascending Chirp plus Gaussian noise, (*Bottom Right*) Descending Chirp plus Gaussian noise, i.e., class 1. The samples contained in the training and testing set are higher in frequency and close to the Nyquist frequency.

#### C.1.2 FILTERS

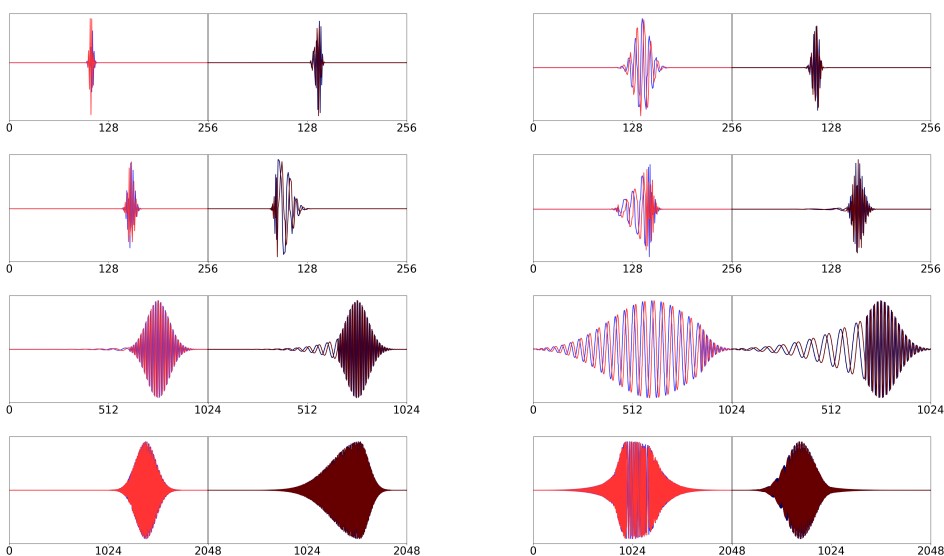

Figure 8: **Learnable Group Transform Filters** for the Artificial Data - Each row displays two selected filters (left and right sub-figure) for different settings: (*from top to bottom*) LGT, nLGT, cLGT, cnLGT. For each subfigure, the left part corresponds to the filter before training and the right part to the filter after training. The blue and red denote respectively the real and imaginary part of the filters.

### C.1.3 GROUP TRANSFORM

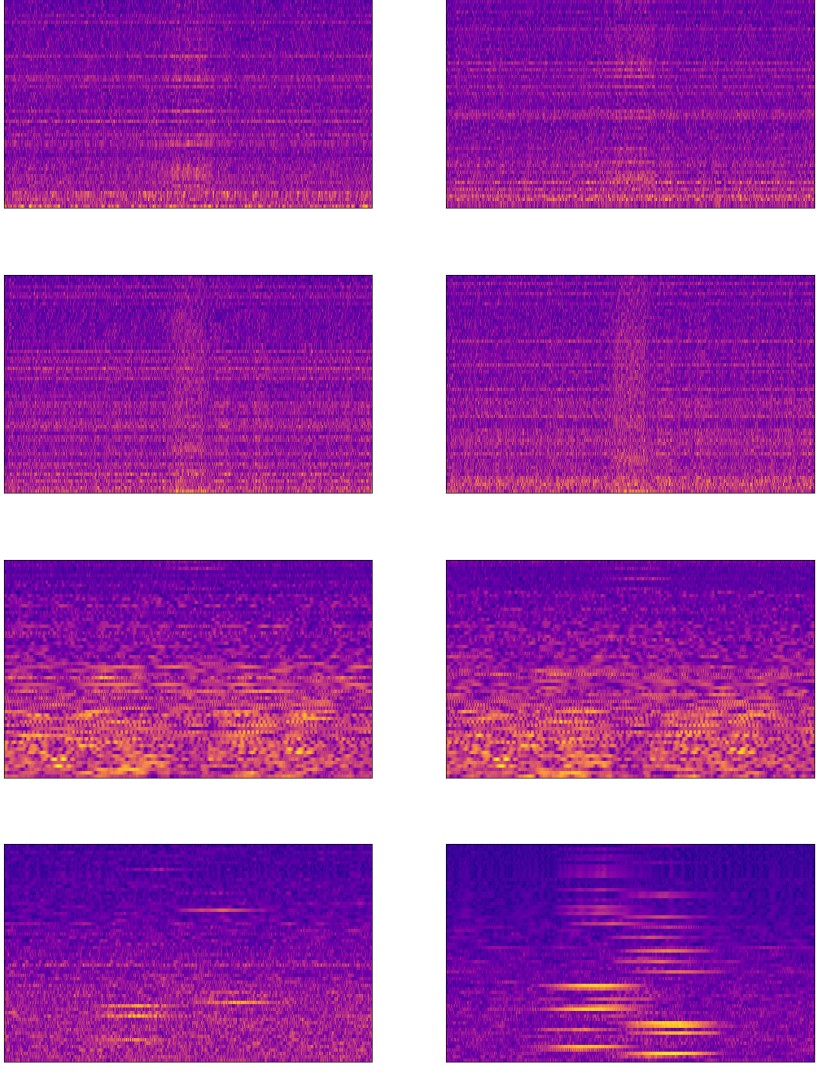

Figure 9: **Learnable Group Transform** - Visualisation of an ascending chirp sample, where for each row (*left*) at the initialization and (*right*) after learning. Each row displays a different setting: (*from top to bottom*): LGT, nLGT, cLGT, cnLGT.

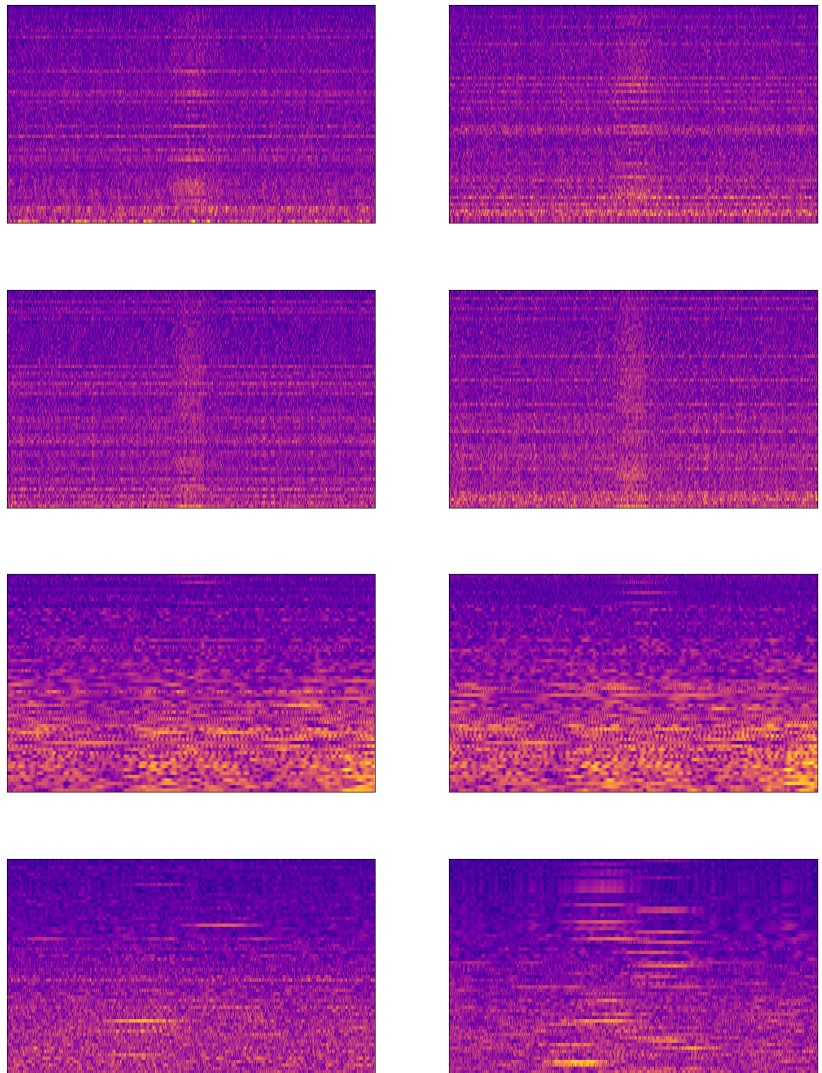

Figure 10: **Learnable Group Transform** - Visualisation of a descending chirp sample, where for each row (*left*) at the initialization and (*right*) after learning. Each row displays a different setting: (*from top to bottom*): LGT, nLGT, cLGT, cnLGT.

## C.2   SUPERVISED BIRD DETECTION

### C.2.1   DATA

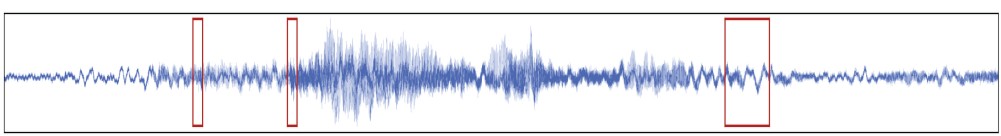

Figure 11: **Bird Detection Dataset** - Sample containing a bird song. The red boxes are the locations of the bird song.

Each data sample, normalized, centered and subsampled by two before experiment.

## C.2.2 FILTERS

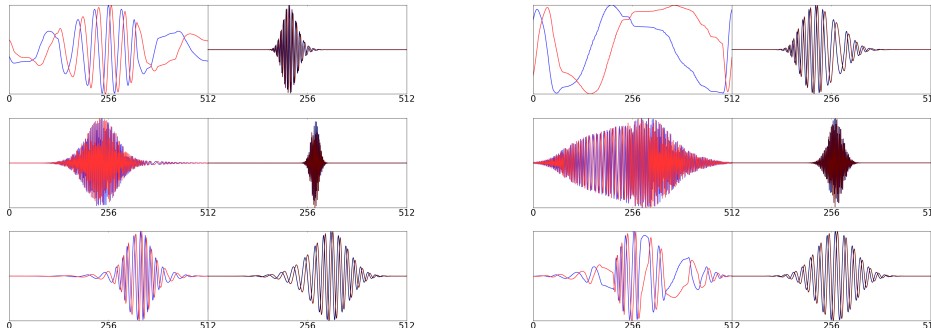

Figure 12: **Learnable Group Transform Filters** for the Bird Detection Data - Each row displays two selected filters (left and right sub-figure) for different settings: (*from top to bottom*) LGT, nLGT, cLGT. For each subfigure, the left part corresponds to the filter before training and the right part to the filter after training.

## C.2.3 GROUP TRANSFORM

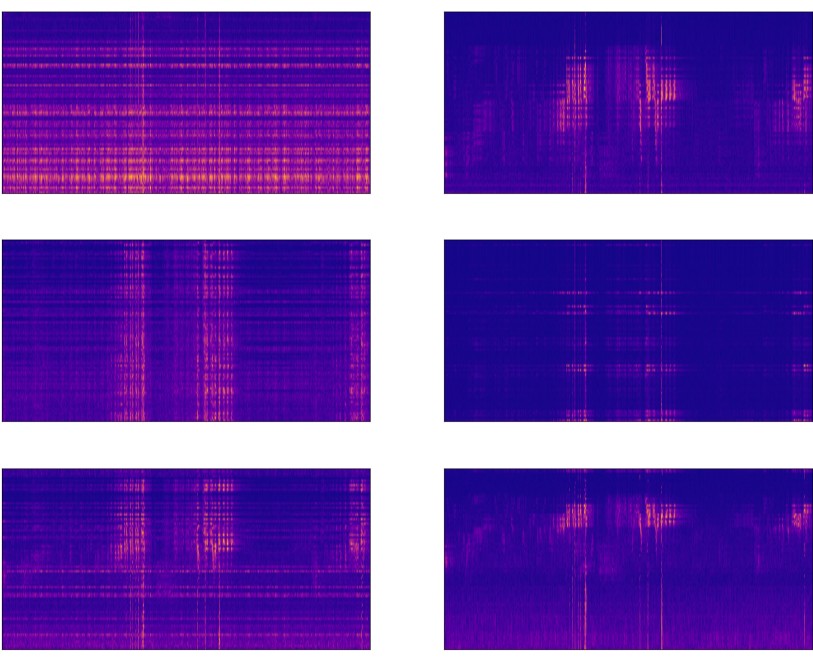

Figure 13: **Learnable Group Transform** - Visualisation of a sample containing a bird song, where for each row (*left*) at the initialization and (*right*) after learning. Each row displays a different setting: (*from top to bottom*): LGT, nLGT, cLGT.

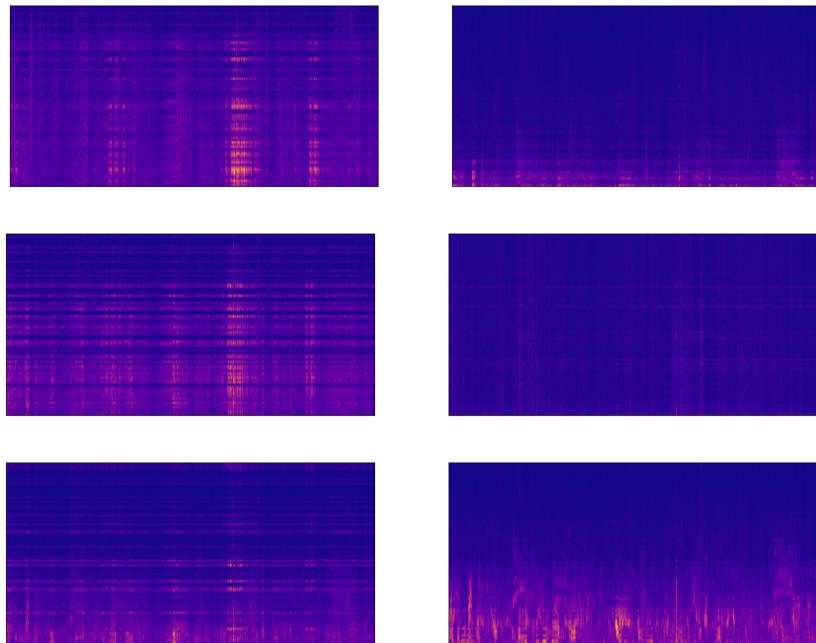

Figure 14: **Learnable Group Transform** - Visualisation of a sample without a bird song, where for each row (*left*) at the initialization and (*right*) after learning. Each row displays a different setting: (*from top to bottom*): LGT, nLGT, cLGT.

## C.3 HAPTICS DATA

### C.3.1 DATA

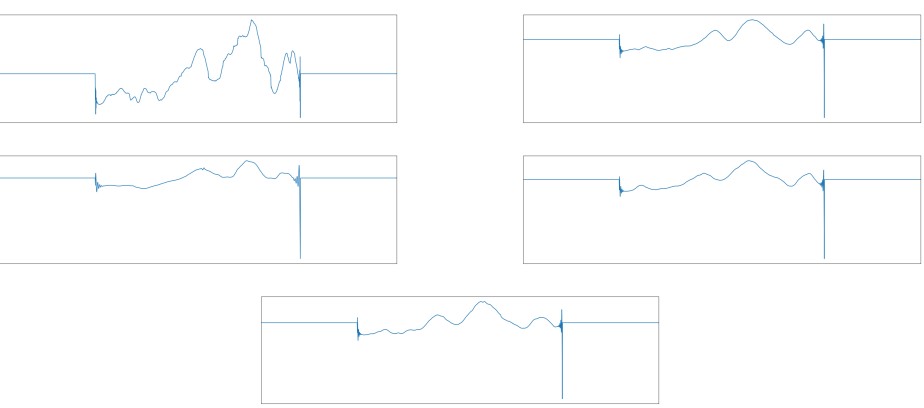

Figure 15: **Haptic Dataset** - Sample of each class of the Haptic dataset.

Each data is centered and normalized. For the experiments, the number of epochs is set to $1000$ and we perform early-stopping and obtain the testing accuracy at this specific epoch as in Khan & Yener (2018), the batch size was set to $64$. In order to avoid overfitting, we perform different asymmetric zeros-paddings on the training samples. For the testing samples, we perform a symmetric zeros-padding ($512$ zeros on each side of the signals).

### C.3.2 FILTERS

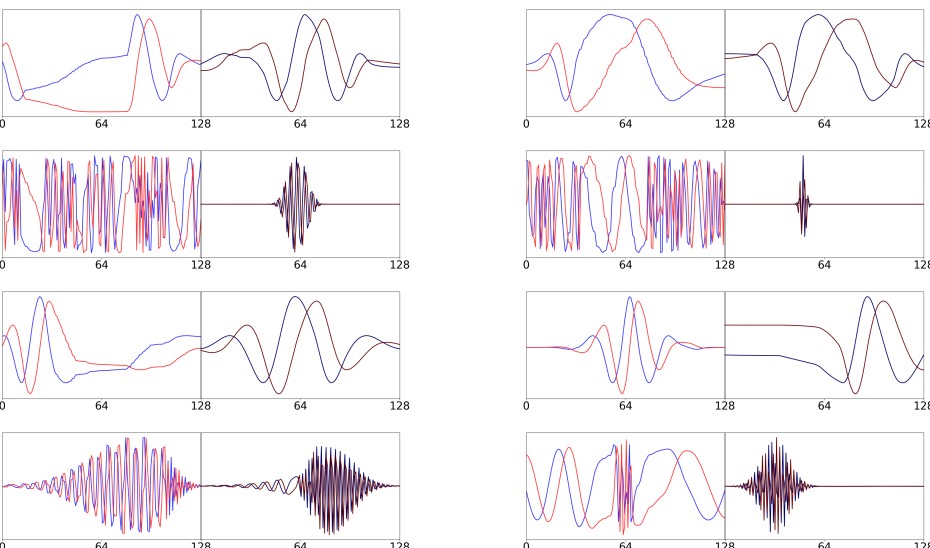

Figure 16: **Learnable Group Transform Filters** for the Haptics Data - Each row displays two selected filters (left and right sub-figure) for different settings: (*from top to bottom*) LGT, nLGT, cLGT, cnLGT. For each subfigure, the left part corresponds to the filter before training and the right part to the filter after training. The blue and red denote respectively the real and imaginary part of the filters.

### C.3.3 GROUP TRANSFORM

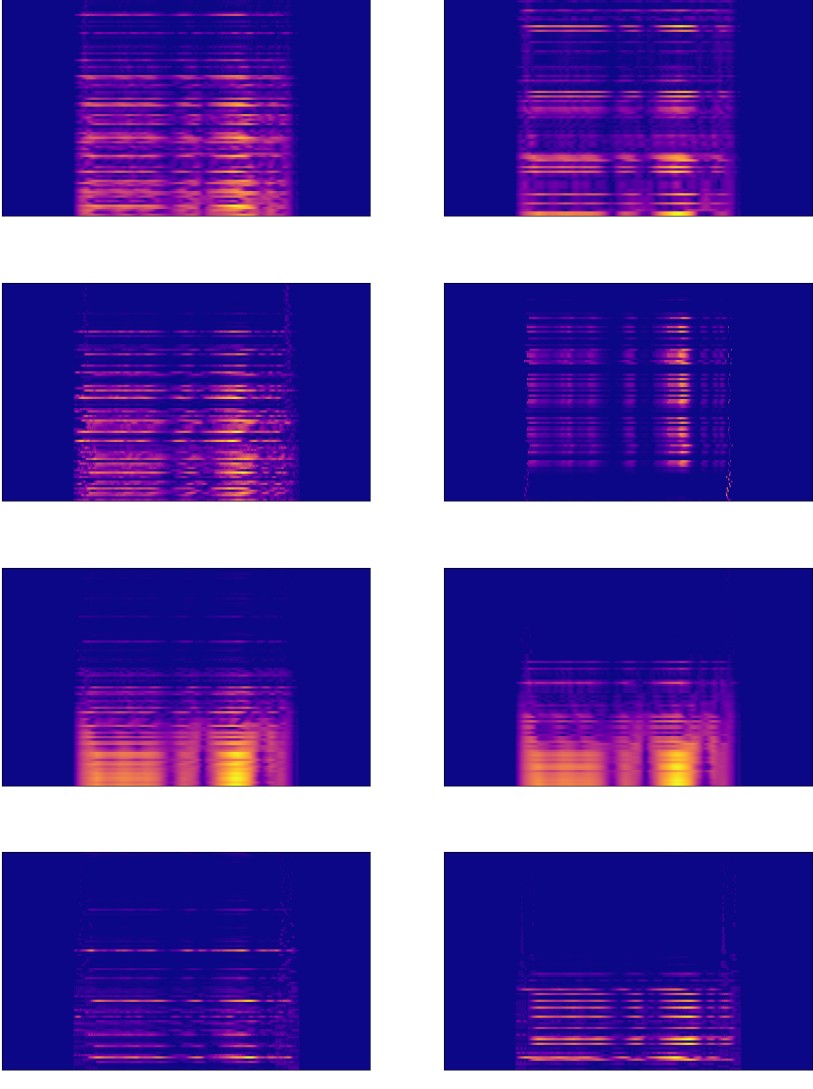

Figure 17: **Learnable Group Transform** - Visualisation of a sample belonging to class 1, where for each row (*left*) at the initialization and (*right*) after learning. Each row displays a different setting: (*from top to bottom*): LGT, nLGT, cLGT, cnLGT.

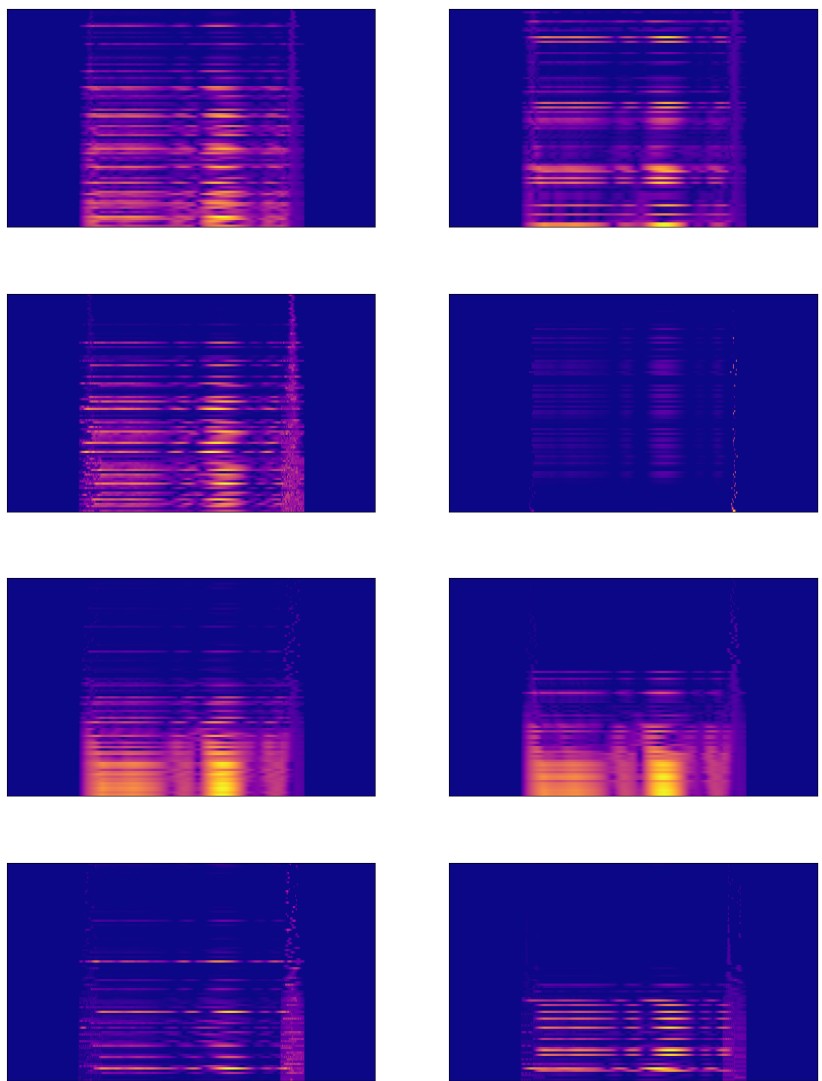

Figure 18: **Learnable Group Transform** - Visualisation of sample belonging to class 2, where for each row (*left*) at the initialization and (*right*) after learning. Each row displays a different setting: (*from top to bottom*): LGT, nLGT, cLGT, cnLGT.

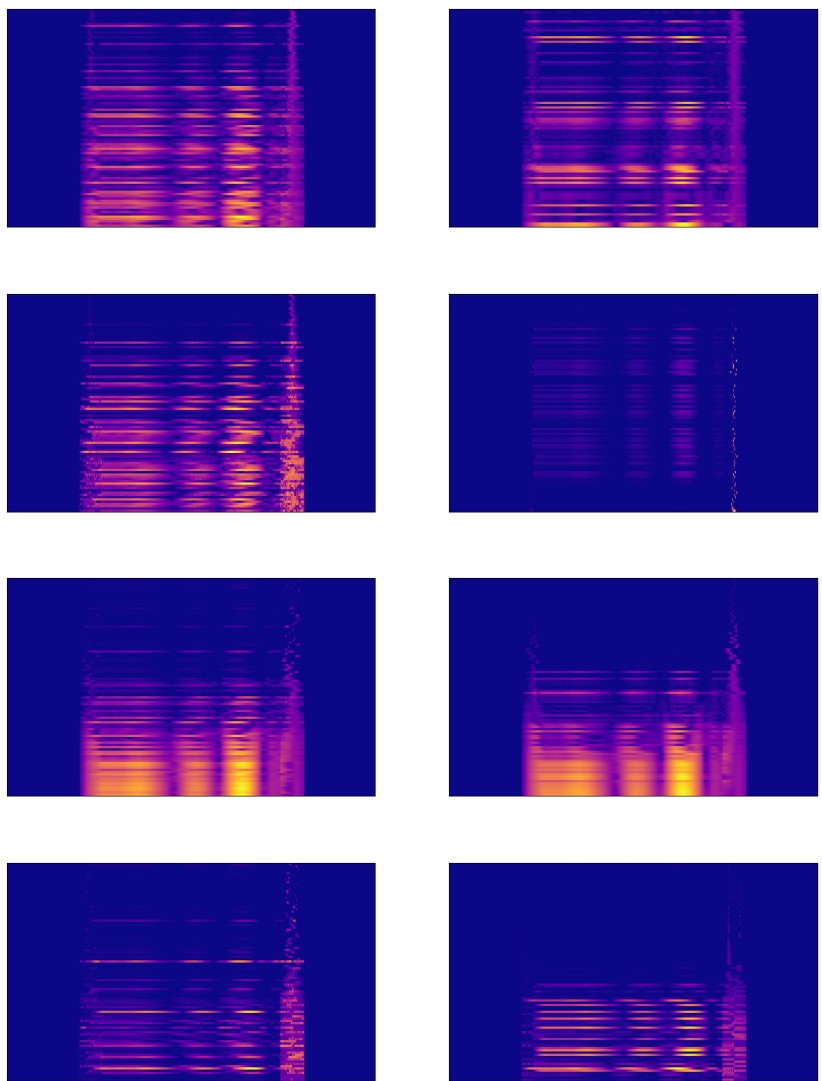

Figure 19: **Learnable Group Transform** - Visualisation of sample belonging to class 3, where for each row (*left*) at the initialization and (*right*) after learning. Each row displays a different setting: (*from top to bottom*): LGT, nLGT, cLGT, cnLGT.

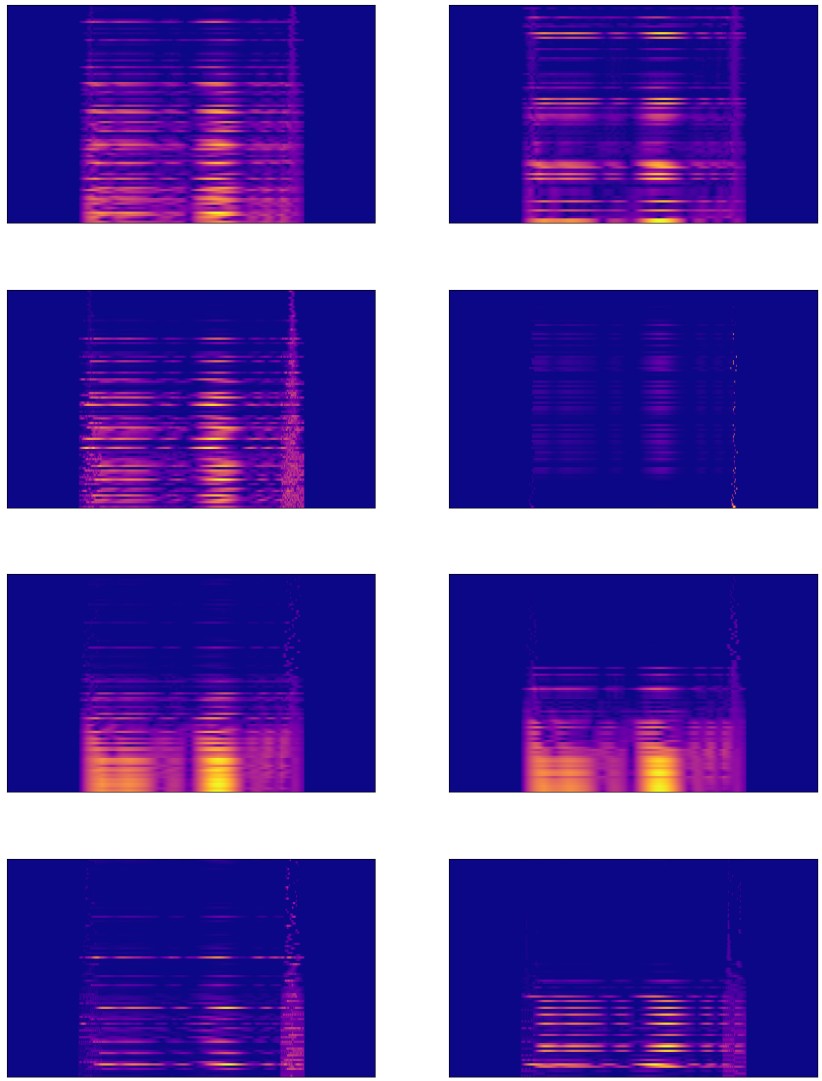

Figure 20: **Learnable Group Transform** - Visualisation of a sample belonging to class 4, where for each row (*left*) at the initialization and (*right*) after learning. Each row displays a different setting: (*from top to bottom*): LGT, nLGT, cLGT, cnLGT.

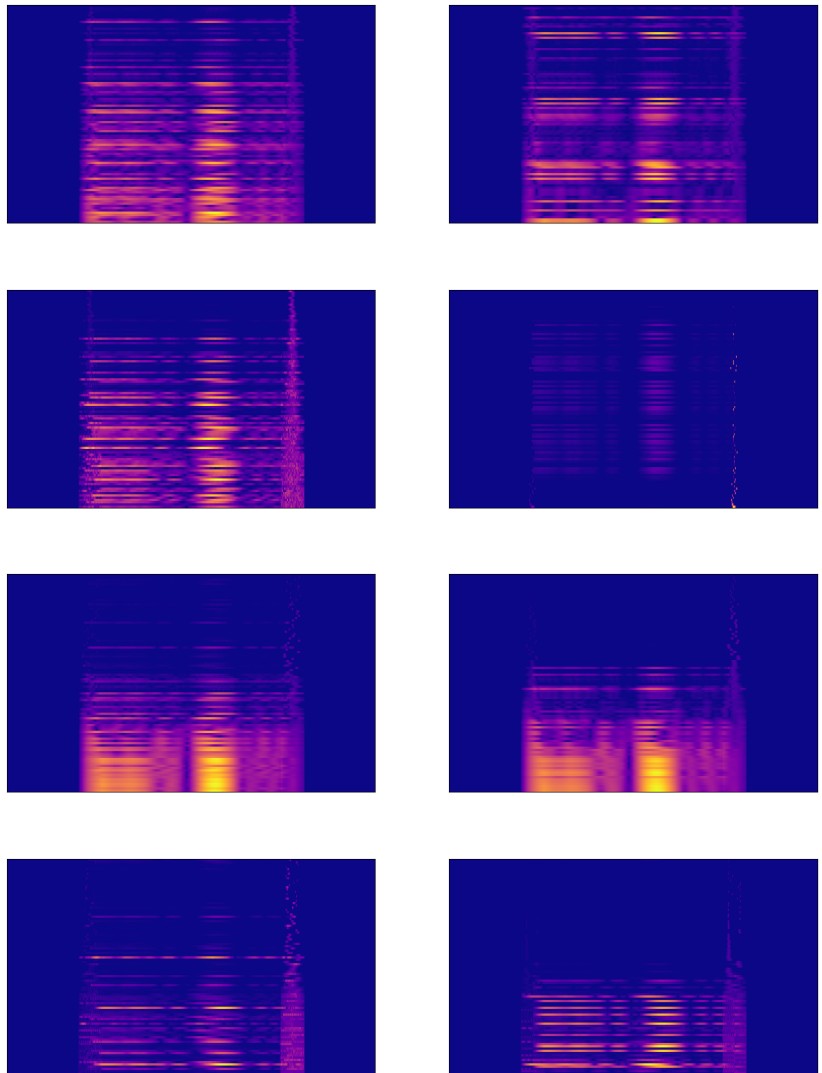

Figure 21: **Learnable Group Transform** - Visualisation of a sample belonging to class 5, where for each row (*left*) at the initialization and (*right*) after learning. Each row displays a different setting: (*from top to bottom*): LGT, nLGT, cLGT, cnLGT.

## D    GROUP PARAMETER OPTIMIZATION

In order to learn the group transform module, we can use the back-propagation algorithm and a gradient-based optimization technique such that the parameters of the group transform module, denoted by $g$, can be learned jointly with the parameters of the DNN, or any other differentiable algorithm taking as input the learnable time-frequency representation. Using the notations of Section 3.3 where $L$ denotes a loss function and $F$ a DNN, the learnability of the optimal group transform leading to the most suitable time-frequency representation is performed by the chain rule,

$$\frac{\partial L}{\partial g_k} = \frac{\partial L}{\partial [F(\mathcal{W}_\psi(g_k, s_i))]} \times \frac{\partial [F(\mathcal{W}_\psi(g_k, s_i))]}{\partial g_k}, \forall i \in \{1, \ldots, N\}, \forall k \in \{1, \ldots, K\}, g_k \in \mathbf{G}_{\text{inc}},$$

where $[\mathcal{W}_\psi(g_k, s_i)]$ is the convolution of the signals $s_i$ with the transformed filter $\rho_{\text{inc}}(g_k)\psi$ as defined in (14).

# E PROOFS

**Proposition 2.** $\rho_{inc}$ *is a group representation of* $\boldsymbol{G}_{inc}$ *on* $\mathbb{L}_2(\mathbb{R})$.

## E.1 PROOF THEOREM 2

*Proof.* Let $g, g' \in \mathbf{G}_{\mathrm{inc}}$, then

$$[\rho_{\mathrm{inc}}(g' \circledast g)\psi](t) = \psi((g' \circledast g)(t))$$
$$= \psi(g'(g(t)))$$

and,

$$[\rho_{\mathrm{inc}}(g')\rho_{\mathrm{inc}}(g)\psi](t) = [\rho_{\mathrm{inc}}(g')\psi](g(t))$$
$$= \psi(g'(g(t)))$$

which verifies the homogeneity property. The linearity is implied by,

$$[\rho_{\mathrm{inc}}(g)(\kappa\psi_1 + \psi_2)](t) = (\kappa\psi_1 + \psi_2)(g(t)) = \kappa\psi_1(g(t)) + \psi_2(g(t)), \forall t \in \mathbb{R}.$$

where $\psi_1, \psi_2 \in \mathbb{L}_2(\mathbb{R})$ and $\kappa \in \mathbb{R}$. It is in fact a Koopman operator Korda & Mezić (2018). $\square$

## E.2 PROOF THEOREM 1

*Proof.* Let $\tau \in \mathbb{R}$ and $g, g' \in \mathbf{G}_{\mathrm{inc}}$,

$$\mathcal{W}[\rho_{\mathrm{inc}}(g')s_i, \psi](g, \tau) = \langle \rho_{\mathrm{inc}}(g')s_i, \rho_{\mathrm{inc}}(g)\psi_\tau \rangle$$
$$= \langle s_i, \rho_{\mathrm{inc}}(g')^{-1}\rho_{\mathrm{inc}}(g)\psi_\tau \rangle$$
$$= \langle s_i, \rho_{\mathrm{inc}}(g'^{-1})\rho_{\mathrm{inc}}(g)\psi_\tau \rangle$$
$$= \langle s_i, \rho_{\mathrm{inc}}(g'^{-1} \odot g)\psi_\tau \rangle$$
$$= \mathcal{W}[s_i, \psi](g'^{-1} \odot g, \tau),$$

where $\psi_\tau$ denotes the filter $\psi$ centered at position $\tau$. Then, there is not guarantee that this can be extrapolated to all $\tau \in \mathbb{R}$, i.e., in the convolution case, except in the affine case where the global transformation matches the iteration of a local one.

$\square$

