# OpenReview forum: "Learnable Group Transform For Time-Series"
_ICLR.cc/2020/Conference — Reject_

### Official Review · AnonReviewer1 · 2019-10-22
**Official Blind Review #1**

**Rating:** 3

**Review:**

This paper defines a set of learnable basis functions and a joint learning algorithm to estimate them. It is based on the premise that the common learning approach in time-series is to first represent them in some spectral domain; thus, the main problem is to define and estimate the basis functions. However, this premise is not accurate and many learning algorithms operate just in the actual time domain (even in speech).

The paper advocates for a special group of strictly increasing transformations of the basis functions. While in Section 3.2 the authors do explain the connections to the time-warping operation, they never explicitly justify why such a group is interesting. At least, it is not clear why they are not interested in a semi-group of (non-strictly) increasing functions, which is more flexible and used in DTW.

The authors implement the more general group using piece-wise linear functions, which makes enforcing the strictly increasing property easier. This idea is nice, however I am concerned with speed of such transformations in practice. Unfortunately the authors do not report runtime numbers in the experiments.

While the paper has verbose and wordy writing, it also spends a major part of the paper describing the common knowledge about group theory and wavelets. Instead, some crucial design decisions are not well-justified. For example, the description of the constraints in cLGT suddenly appears.

There are several major problems in the experiments. First, the results in Tables 1 and 2 do not seem to be statistically significant and Table 3 does not have the intervals. Second, the Haptics dataset is too small. Also, choosing a single dataset from the UCR sets only sends the message that this dataset is one of the few datasets on which the algorithm has worked better.
Overall, the key message out of the experiments is that if you project the data to a parameterized basis set and constrain the parameters, you will get better generalization, which is not very novel.

Other than improving the experiments, the authors can significantly improve the writing by decreasing the emphasis on the background knowledge and elaborating more on the new proposed ideas.

**Experience Assessment:**

I have read many papers in this area.

**Review Assessment: Checking Correctness Of Derivations And Theory:**

I assessed the sensibility of the derivations and theory.

**Review Assessment: Checking Correctness Of Experiments:**

I assessed the sensibility of the experiments.

**Review Assessment: Thoroughness In Paper Reading:**

I read the paper at least twice and used my best judgement in assessing the paper.

---

> ### Author Response · Authors · 2019-11-12
> **Response to Review#1**
>
> We thank the reviewer for his/her feedback. In the following, we answer each point separately, and we updated the paper accordingly.
>
> ”””While in Section 3.2 the authors do explain the connections to the time-warping operation, they never explicitly justify why such a group is interesting.”””
>
>     ---> The last paragraph of the introduction is dedicated to such justification. We also added:
>
> This specific transformation of a filter induces a non linear transformation of the instantaneous phase of the filter allowing to span filters a la chirplets. These filters are of interest in a variety of domains such as biology and medicine, mechanics and vibrations, and sonar systems [3], and especially in bird song.
>
> ”””At least, it is not clear why they are not interested in a semi-group of (non-strictly) increasing functions, which is more flexible and used in DTW.”””
>
>     ---> As we can see from the experiments, for some cases, the LGT model is too flexible, and adding some constraints (nLGT,cLGT,cnLGT) on the transform helps to increase the accuracy. As such, more flexibility might require even more constraints on the model, and thus we did not consider it here.
>
> ”””The authors implement the more general group using piece-wise linear functions, which makes enforcing the strictly increasing property easier. This idea is nice, however I am concerned with speed of such transformations in practice. Unfortunately the authors do not report runtime numbers in the experiments.”””
>
>     ---> In practice, a piecewise linear function can be implemented by a single layer Neural Network with ReLU units. As such, the training can be achieved efficiently.
>
> ”””While the paper has verbose and wordy writing, it also spends a major part of the paper describing the common knowledge about group theory and wavelets.”””
>
>     ---> In the updated version, we simplified some of the writings and reduced the background section.
>
> ”””Instead, some crucial design decisions are not well-justified. For example, the description of the constraints in cLGT suddenly appears.”””
>
>     ---> Thanks for the comment. We adjusted the revised paper, and the justifications of the different constraints can be found in section 3.4.
>
> ”””There are several major problems in the experiments. First, the results in Tables 1 and 2 do not seem to be statistically significant and Table 3 does not have the intervals.”””
>
>     ---> For the benchmarks, we proposed to compare it with already existing methods. To do so, we use the same experimental and evaluation procedures as the original published works in NIPS2018 and ICML2018. Also, we do not claim to have a statistically significant improvement across the dataset as the experiments are achieved in order to perform a model comparison. As such, in Table 3, there are no intervals since the method applied for the different benchmarks uses early stopping on the valid set.
>
> ”””Second, the Haptics dataset is too small. Also, choosing a single dataset from the UCR sets only sends the message that this dataset is one of the few datasets on which the algorithm has worked better.”””
>
>     ---> We recall that the bird detection dataset contains about 20 hours of audio signal. As such, the bird detection task aims at comparing it with a learnable mother filter method (ICML2018) and the UCR dataset with the article that is the closest to our work (NIPS2018). Finally, the toy example is here to build intuitions. For each dataset, there is a specific level of knowledge regarding the filters: 1) (bird) large dataset with expert knowledge, 2) (UCR) small dataset with no expert knowledge, 3) (toy example) small dataset with perfect knowledge. Thus, we believe that these three datasets are sufficient to compare and show the efficiency of the developed algorithm.
>
> ”””Other than improving the experiments, the authors can significantly improve the writing by decreasing the emphasis on the background knowledge and elaborating more on the new proposed ideas.”””
>
>     ---> In the updated version, we reduced the emphasis on the background knowledge, especially we moved the wavelet example in the appendix, and offered more analysis regarding the method and its results.
>
> [3] P. Flandrin. Time frequency and chirps. In Wavelet Applications VIII,

---

### Official Review · AnonReviewer3 · 2019-10-22
**Official Blind Review #3**

**Rating:** 8

**Review:**

*Paper summary*

The authors design a learnable time-series pre-processing, which they refer to as a learnable group transform (LGT). This is a generalization of the wavelet transform, which maps a time-series signal onto the affine group. In the wavelet transform, multiple scaled and shifted versions of a mother wavelet are "inner-producted" with a signal; the resulting coefficents are the output of the transform. In the LGT, a more flexible transform that just scaling and shifting is applied to the shape of the mother wavelet, which is piece-wise linearly stretched. This elegantly encompasses time-warping and many other wavelet-style transforms into one learnable preprocessing step.

*Paper decision*

I would like to recommend this paper be accepted. It is clearly written and the idea is simple; that said, the idea is an elegant generalization of the wavelet transform. (I admit my own expertise is not in time-series data, so I may be mistaken). The experiments are also simple, but straightforward and easy to reimplement.

*Supporting arguments*

What I like about this paper is that the idea is a straightforward generalization of the wavelet transform through the lens of group theory. Furthermore, other transforms, such as the short-time fourier transform, or methods such as time-warping are easily encompassed by the method. By making the transform learnable, the authors reduce the number of in-built assumptions in the problem.

I believe the idea is novel; although, since this is not my area of expertise, I defer to other reviewers who may wish to contest this.

I do wonder, however, what the interpretation is behind some of the learned LGTs. Typically, a group transform is favored because of certain symmetry properties of the task at hand. For instance, wavelet transforms are shift-covariant, reflecting the shift-covariance of underlying signal statistics.

Experimentally, I think there was a nice selection of toy and harder examples. I found the visualized filters and group transforms in the appendix really interesting. I think it would have been nicer to see some more analysis of the interpretation of the learned LGT in the main text though. Further, it would have been nicer to see some ablation studies conducted, such as varying the number of degrees of freedom in the LGT and seeing how that affects performance.


*Smaller questions/notes for the authors*

- The explanation of what a group is is quite high-level and I think if you did not know what it was beforehand, it would be hard to understand what one is. For instance, just after introducing groups, homomorphisms are mentioned. I would guess that someone who has never seen a group beforehand, would have no idea about homomorphisms.

- In equation 6, is \star a convolution or a cross-correlation? It looks like a cross-correlation to me.

- I really liked the connection drawn between the group transform and time-warping. This is an aspect I personally had never considered.

- Would it be possible to move some of the material on the learned filter transformations to the main text and write some analysis, even if it is only qualitative? I would find that fascinating.


**Experience Assessment:**

I have published one or two papers in this area.

**Review Assessment: Checking Correctness Of Derivations And Theory:**

I carefully checked the derivations and theory.

**Review Assessment: Checking Correctness Of Experiments:**

I assessed the sensibility of the experiments.

**Review Assessment: Thoroughness In Paper Reading:**

I read the paper thoroughly.

---

> ### Author Response · Authors · 2019-11-12
> **Reponse to Review #3**
>
> We thank the reviewer for his/her feedback. In the following, we answer each point separately, and we updated the paper accordingly.
>
> """I do wonder, however, what the interpretation is behind some of the learned LGTs(...)”””
>
>     ---> For any learned transformation, we use a convolution of the filter with the signal; it is thus a shift-covariant transformation. Besides, wavelet transforms are dilation-covariants. In our case, if the transformation learned is affine, we recover such covariance. However, if the transformation is non-linear, we are losing this covariance property. We updated the paper to make these points clear in section 3.5 Group Equivariance.
>
> ”””Experimentally, I think there was a nice selection of toy and harder examples (...) more analysis of the interpretation of the learned LGT in the main text though.”””
>
>     ---> Thank you for this input, we updated the paper accordingly.
>
> ”””The explanation of what a group is is quite high-level (...), homomorphisms are mentioned. I would guess that someone who has never seen a group beforehand, would have no idea about homomorphisms.”””
>
>     ---> We have revised the paper accordingly (sec 2.2) to make this point more intuitive and reduced the emphasis on the homomorphism.
>
> ”””In equation 6, is \star a convolution or a cross-correlation? It looks like a cross-correlation to me.”””
>
>     ---> We updated the notations so that it corresponds to a convolution
>
> ”””Would it be possible to move some of the material on the learned filter transformations to the main text and write some analysis, even if it is only qualitative? I would find that fascinating.”””
>
>     ---> We proposed a qualitative analysis of Figure 4 as well as includes more filters for the bird detection experiment. For the filters, we offer these qualitative aspects:
>          One can notice that all the learned filters in Figure 5 contain either an increasing chirp or a decreasing chirp, corresponding respectively to the convexity or concavity of the instantaneous phase of the filter and thus of the piece-wise linear map. Such a feature is being used and is crucial in the detection and analysis of bird song [2].
>
> [2] D. Stowell and M. D Plumbley. Framewise heterodyne chirp analysis of birdsong.

---

### Official Review · AnonReviewer2 · 2019-10-23
**Official Blind Review #2**

**Rating:** 3

**Review:**

A typical Wavelet Transform is built through the dilation and/or rotation of a mother wavelet, which can been viewed as a group action on a mother wavelet. This work proposes to extend this construction beyond the Euclidean group, and to supervisedly learn operators that will be applied on a mother wavelet. Competitive numerical performances are obtained.

Overall, I think that re-thinking the way a Wavelet Transform is designed, is an interesting direction of research, but I think some of the theoretical tools developed in this paper are not dedicated to achieve this purpose. In particular, the group/representation properties seem to not be used, and the authors could simply consider a specific subset of invertible mapping on $\mathbb{R}^2$ which would be applied on the mother wavelet and lead to a Wavelet Transform. In other words, the overall formulation could be simplified.


Pros:
- In general, the numerical experiments are at the level of the state of the art.
- Parametrizing a subset of the group of increasing function and its application to signal processing tools is novel, to my knowledge.

Cons:
- Some very relevant elements in the literature review are missing. Learning or using an underlying group of symmetry that will be combined with a deep neural network is not novel, cf: https://arxiv.org/abs/1601.04920 ; in particular for reducing the number of parameters, filters or samples: https://arxiv.org/abs/1809.06367 ; http://proceedings.mlr.press/v48/cohenc16.pdf ; https://arxiv.org/abs/1809.10200 ; https://arxiv.org/abs/1605.06644 - I think the authors should discuss at least one or two of those papers, if not all.
- The performance on the bird detection task is good but the improvement compared to other work is not clear, given that some supervision in the first layer is incorporated.
- Subsections 2.2 and 2.3 are difficult to parse because the authors introduce a lot of equations or notion that are not useful to understand their algorithm/method. The equation (6) seems wrong to me (one should consider t->s_i(-t) and not t->s_i(t) and b seems missing in the second line).
- Figure 2 is difficult to read because of the illustrative graphics. Maybe a block schema would be easier to parse.
- It seems to me that no-where the group properties are used, such as the stability to composition. In this paper, the authors simply try to parametrize a diffeomorphism to dilate the mother wavelet. From my understanding of 3.3, the subset of function used to approximate $G_{inc}$ do not form a subgroup as well, contrary to the Euclidean case, where for instance discrete rotations in the case of images are a finite group.
- In subsection 4.1(Table 1), a comparison with a wavelet transform followed by a linear operator is compared with the proposed method. I find this result surprising : LGT/nLGT/cLGT/cnLGT and the WT are some linear methods whereas the STFT is non linear. As the WT should be unitary, if the linear classifier method is reasonably trained, then both methods should lead to the same result, except if the data are poorly conditioned. In which case, this experiment would not be meaningful. I think the authors should comment more this result because it is surprising.
- I slightly disagree with the sentence "in the case of WT, the precision in frequency degrades as the frequency increases". Actually, the heisenberg principle is optimally optimized by wavelets, meaning that the area of the frequency/spatial support on a spectrogram is constant. On the contrary, the STFT has a lack of localisation (and thus the "precision" is not constant along frequencies). Maybe this could be rephrased slightly.
- Given the filter learned in Figure 5, one can wonder if a foveal approach (i.e., Foveal Wavelets) could perform similarly? It would be interesting to display the littlewood-paley plot(i.e., the sum of the modulus of the filters in the Fourier domain) of this representation to understand the nature of this operator in the Fourier domain.
- I think group actions could be considered instead of representations: it would be simpler to understand for a potential reader.

Typos:
- abstract "in order to transform the mother .." > "in order to transform a mother.."
- page 7 "this variation is as not captured as well" > "this variation is not captured as well".


**Experience Assessment:**

I have published in this field for several years.

**Review Assessment: Checking Correctness Of Derivations And Theory:**

I carefully checked the derivations and theory.

**Review Assessment: Checking Correctness Of Experiments:**

I carefully checked the experiments.

**Review Assessment: Thoroughness In Paper Reading:**

I read the paper thoroughly.

---

> ### Author Response · Authors · 2019-11-12
> **Response to review#2**
>
> We thank the reviewer for his/her feedback. In the following, we answer each point separately, and we updated the paper accordingly.
>
> Answers to the "Cons" section:
>
> 1) We added in Section 3 the equivariance property of the group where we make use of the representation formulation. Besides, our formulation is required to develop theoretical insights regarding these learnable filters, such as the Haar measure (although in our case, it should be a quasi-invariant measure instead of an invariant). From this measure, we can look for constraints to build a resolution of the identity. Recently, the theoretical study of such a non-uniform filter bank has been investigated in [1] from a frame theory approach. Our framework enables the development of such a theory from a coherent state (representation theory) point of view. We made this clear in Section 2.
>
> 2) We added some of the papers you mentioned in the new paragraph on the equivariance properties of the group in Section 3.
>
> 3) We performed the experiments with four different settings. We updated the paper to emphasize the following points:
> The case without constraints (LGT) reaches better accuracy than the domain expert benchmark (MFSC).
> Including more constraints on the model (cnLGT) reduces overfitting and further improves the results to outperform the other benchmarks.
>
> 4) Thanks for your input on this, it has been corrected in the updated version.
>
> 5) We updated Figure 2 to improve its clarity.
>
>  6) As we mentioned, we do not propose to approximate G_inc but to sample it. Increasing piecewise linear functions do not form a group under the composition operation. In the same way that WT uses the dyadic sequence $\left \{1,\dots,2^{j/Q}, j \in \left \{0,\dots, J \times Q -1 \right \} \right \}$ to sample the dilation parameter of the affine group while this discretization does not form a group under the multiplication operation.
>
> 7)        -As you know, STFT is a linear time-frequency transform as opposed to the Spectrogram which is the square modulus of the STFT. Please refer to A wavelet tour of signal processing, S. Mallat, for more details.
>
>             -We are not clear about your point. All those different groups transform correspond to representing the data in a new “basis,” then each new “basis” will impact the linear separability of the data and thus the linear classifier performances.
>
> 8)        -The Heisenberg uncertainty principle is optimal for the Gabor    transform, which is a special case of STFT with a Gaussian window. Then for the wavelet transform, the area of each tile will depend on the wavelet family and parameterization.
>           -As you know, in the case of WT, the area of the tiles remains the same, but the resolution in time and frequency depends on the dilation parameter. That is, the constant ratio bandwidth to center frequency of the wavelet filters implies that high-frequency filters will have “poor” frequency resolution and “high” time resolution. For STFT, the window is the same across the time-frequency plane leading to filters with the same time and frequency resolution. We updated the paper to make that clear.
>
> 9) We focused on giving insights from the group theory formulation as it is one of the primary building blocks to create a time-frequency transform. Besides, as we believe that there is no paper in the machine learning community discussing this approach for learnable time-frequency transformation, it is essential to explain it in detail. We updated the article to provide more insights on the experiments from a signal processing point of view.
>
> 10) The typos have been corrected.
>
> [1] Continuous warped time-frequency representations - Coorbit spaces and discretization, N. Holighaus et al.

---

> > ### Comment · AnonReviewer2 · 2019-11-13
> > **Thanks for the precisions**
> >
> > Dear authors,
> >
> > Thanks, I've also read your revisions via the pdf comparisons. Some points are addressed.
> >
> > 1) Again, the method which is developed in this paper can be summarized as: a/ parametrize a subset of the diffeomorphisms b/ consider its action on some mother wavelet(s) c/ learn the parameters. I do not understand which theoretical properties "group representations" bring, and this rebuttal didn't help me much to understand.
> >
> > 3) Adding one reference sounds fine.
> >
> > 4)5) Thanks.
> >
> > 6) This is what I understood. Then, clearly, the group structure is not used in this "theory". The case of the dilation is specific: the action of scales \{j\in \mathbb{N}\}  on mother wavelets is a group, but due to aliasing issues, one has to sample them more. This indeed has no connexion with "group theory" but this is classical.
> >
> > 7) Are you really stating that linear separability can be improved by a linear operator? Thanks for the reference about the STFT(I'm used to the name "windows fourier transform"), indeed, I got confused that one compares linear representations with a linear classifier. Am I misunderstanding something? I would appreciate a clarification.
> >
> > 8) Thanks. All I meant, again, is that usually the adaptive bandwidth property of dilated wavelets is better suited for describing low/high frequencies structure of a signal, compared to the windows fourier transform for which a trade-off has to be done. Your sentence still suggests that this is a disadvantage of the wavelets transform, whereas I claim this is often an advantage... (page 93/109 of the aforementioned book)
> >
> > 9) Cf 1). What are those "insights"? While I think this approach is interesting, I still believe it could be widely simplified.

---

> > > ### Author Response · Authors · 2019-11-13
> > > **Reponse to review #2 (2)**
> > >
> > > We thank you for your prompt reply and interest.
> > >
> > > 1) 9)  Our approach aims at building a recipe allowing the learnability of the samples of a group governing the filterbank’s orbit and derive its equivariance property. It is clear that this approach can be directly used, for instance, to develop learnable invariant representations a la Scattering Network by taking an averaging operator (or a max) over the samples of the group. Because of the representation theory formulation, we know that the filters are sampled along an orbit, and we know which point of such orbit has been sampled (via the parameters $(a_k,b_k)$). We can thus characterize the invariance by considering which sample has been learned on the orbit.
> > >
> > > 7)  In our setting, it was the linear transformation followed by a non-linearity. We meant a change of basis before the non-linearity. Without the non-linearity, we obviously agree that it will not alter the classification of the linear classifier.
> > >
> > > 8) We agree that in some cases, this representation (WT) is best suited, as for many biological signals. However, given a signal and a specific task, it is not clear which representation and thus which time-frequency trade-off should be used. It usually requires expert knowledge of the features of interest in the signal or cross-validation. Having an adaptive representation allows alleviating this issue as it is empirically shown in the different experiments and especially in the artificial dataset.

---

> > > > ### Comment · AnonReviewer2 · 2019-11-13
> > > > **Answer**
> > > >
> > > > Dear authors,
> > > >
> > > > Thanks for your reply.
> > > >
> > > > 7) OK. This should be explicit in the text, I think this is missing and extremely confusing. I've just realized it was in the Appendix, but I didn't check until now that a logarithm was incorporated as (to my understanding/knowledge) this is stated no-where and the caption of the Table should have been self-suffisant(it's written "representation+linear classifier") Has the logarithm a bias incorporated, i.e., does it behave as a thresholding? Is the threshold chosen through supervision? Was it critical?
> > > >
> > > > 8) I think I understand this - but standard properties such as the fact to obtain an isometry, to avoid losing signal structures OR the trade-offs in bandwidth seem not really at the core of this paper. Instead, what is claimed(and that you just said) is that the supervision will find itself the best set of hyper-parameters(i.e., some implicit optimality assumption) - doesn't it also discard possible interpretations?
> > > >
> > > > 1)9) A standard wavelet transform isn't built with randomly sampled elements of a group: the set of elements has in general a (finite/discrete) group structure (at least for 1D and 2D), in order to achieve equivariance. Sampling along the orbit of a group to obtain this structure isn't trivial at all or suffisant.(for instance there is no discrete approximation of 3D rotations by a discrete group) Those group structures are often necessary to obtain non-trivial invariants. Thus, I'm not sure that this method is specifically designed for group invariance via the scattering transform... but this is not the point of this paper.
> > > >
> > > > Again, my concern is simply that the vocabulary/framework from group representations is neither fully used in this work nor necessary to describe the results of this paper. Restraining the theory to the action of a parametrized subset of diffeomorphism wouldn't hurt the (numerical and theoretical) results obtained here(in this paper). And there is a significant literature discussing this(cf my original review). Also, I'd like to recall that a group action on a vector space is simply a representation, but I think none of those specificity are used here.

---

> > > > > ### Author Response · Authors · 2019-11-14
> > > > > **Answer**
> > > > >
> > > > > Dear reviewer,
> > > > >
> > > > > Thank you again for your insights and questions.
> > > > >
> > > > > 7)  Thank you for pointing this out. We updated the pdf accordingly. A bias of the order $10^{-3}$  has been added as it is done in [1] in order to compare with them. We then used the same bias for all the experiments.
> > > > >
> > > > >
> > > > > 8) The learnability and flexibility indeed discard interpretations. On the other hand, compared to plain 1d CNN filter, we have filters that, for some experiments, resemble classical time-frequency atoms such as chirplet in the bird detection task.
> > > > >
> > > > > 1)9)  Thank you for having brought into discussion this point. I understand and agree that the group theory section can be restricted and the overall approach simplified. We modified the PDF by restricting the group theoretical aspect to the equivariance section only and introducing more of the references your originally proposed. The other sections of the paper are now simplified and use the invertible transformation map point of view.
> > > > >
> > > > >
> > > > >
> > > > >
> > > > > [1]  Spline Filters For End-to-End Deep Learning, Randall Balestriero, Romain Cosentino, Herve Glotin, Richard Baraniuk

---

> > > > > > ### Comment · AnonReviewer2 · 2019-11-15
> > > > > > **Thanks for your reply**
> > > > > >
> > > > > > 7/ I see, those bias are quite critical for applications. Thanks for the clarification
> > > > > >
> > > > > > 8/ I see. Thanks for the clarification
> > > > > >
> > > > > > 9/ Thanks. I've read carefully the paper and I honestly think the paper is more clear and precise.
> > > > > >
> > > > > > I will reconsider my review during the discussion phase.

---

> > > > > > > ### Author Response · Authors · 2019-11-15
> > > > > > > **Answer**
> > > > > > >
> > > > > > > We thank you for your consideration and great help to improve the paper.

---

### Decision · Program_Chairs · 2019-12-19

**Decision:**

Reject

**Comment:**

This paper received two weak rejects (3) and one accept (8).  In the discussion phase, the paper received significant discussion between the authors and reviewers and internally between the reviewers (which is tremendously appreciated).  In particular, there was a discussion about the novelty of the contribution and ideas (AnonReviewer3 felt that the ideas presented provided an interesting new thought-provoking perspective) and the strength of the empirical results.  None of the reviewers felt really strongly about rejecting and would not argue strongly against acceptance.   However, AnonReviewer3 was not prepared to really champion the paper for acceptance due to a lack of confidence.  Unfortunately, the paper falls just below the bar for acceptance.  Taking the reviewer feedback into account and adding careful new experiments with strong results would make this a much stronger paper for a future submission.